# Fabrication of Graphene/Zinc Oxide Nano-Heterostructure for Hydrogen Sensing

**DOI:** 10.3390/ma14226943

**Published:** 2021-11-17

**Authors:** Yang-Ming Lu, Chi-Feng Tseng, Bing-Yi Lan, Chia-Fen Hsieh

**Affiliations:** Department of Electrical Engineering, National University of Tainan, Tainan 7005, Taiwan; sper999ppp@yahoo.com.tw (C.-F.T.); kh98310323@gmail.com (B.-Y.L.); dongmen0808@yahoo.com.tw (C.-F.H.)

**Keywords:** graphene, zinc oxide, CVD, hydrothermal, sensor

## Abstract

In this study, hydrogen (H_2_) and methane (CH_4_) were used as reactive gases, and chemical vapor deposition (CVD) was used to grow single-layer graphene on a copper foil substrate. The single-layer graphene obtained was transferred to a single-crystal silicon substrate by PMMA transfer technology for the subsequent growth of nano zinc oxide. The characteristics of CVD-deposited graphene were analyzed by a Raman spectrometer, an optical microscope, a four-point probe, and an ultraviolet/visible spectrometer. The sol–gel method was applied to prepare the zinc oxide seed layer film with the spin-coating method, with methanol, zinc acetate, and sodium hydroxide as the precursors for growing ZnO nanostructures. On top of the ZnO seed layer, a one-dimensional zinc oxide nanostructure was grown by a hydrothermal method at 95 °C, using a zinc nitrate and hexamethylenetetramine mixture solution. The characteristics of the nano zinc oxide were analyzed by scanning electron microscope(SEM),x-ray diffractometer(XRD), and Raman spectrometer. The obtained graphene/zinc oxide nano-heterostructure sensor has a sensitivity of 1.06 at a sensing temperature of 205 °C and a concentration of hydrogen as low as 5 ppm, with excellent sensing repeatability. The main reason for this is that the zinc oxide nanostructure has a large specific surface area, and many oxygen vacancy defects exist on its surface. In addition, the P–N heterojunction formed between the n-type zinc oxide and the p-type graphene also contributes to hydrogen sensing.

## 1. Introduction

Due to the improvement of industrial safety awareness, the detection of flammable gases has attracted more and more attention. Among industrial gases, H_2_ is considered a new type of high-efficiency, clean, and inexhaustible energy source. However, H_2_ is a highly flammable gas. As long as its volume ratio in the air is between 4% and 75%, it burns due to its low ignition energy and large flame propagation speed [1]. Because the storage of H_2_ gas is quite dangerous, in order to reduce the risk of use, the development of a high-sensitivity H_2_ sensor is important. In the future, if H_2_ is to be one of the options for renewable energy, the development of low-cost and high-sensitivity H_2_ sensors is crucial.

Recently, metal oxide nanostructures have been widely studied as gas sensors because of their high sensitivity to target gases, such as zinc oxide [2], tungsten oxide [3], and other metal oxides. This is mainly due to the huge surface area and stable chemical properties of oxides. However, these sensors have the disadvantage of high working temperatures, leading to high power consumption. In addition, when they are used in flammable and explosive gas environments, there are concerns about industrial safety. In order to further improve the sensitivity and other important sensing characteristics, the development of heterogeneous oxide nanostructures has been advocated. In previous studies, a large number of different forms of nanostructures have been proposed, each of which exhibits a unique response to a specific gas [4].

Graphene has excellent characteristics, such as high mechanical strength, thermal stability, good electrical conductivity, and high carrier mobility at room temperature. When used in gas sensor applications, graphene shows poor selectivity among different gases, including H_2_ [5]. This problem can be solved by selecting nanomaterials that have good selectivity for the target gas to modify the surface of graphene. This promotes the synthesis of graphene/metal oxide heterogeneous nanostructures, which can be applied to various sensing technologies [6]. This heterogeneous structure sensor exhibits both of the unique characteristic properties of metal oxide nanostructures and graphene, which enables it to exhibit a good gas sensing response at lower operating temperatures.

The pioneering work of this study is reported in the paper: The role of ALD-ZnO seed layers in the growth of ZnO nanorods for hydrogen sensing, published in 2019 [7]. This study used a graphene/zinc oxide nano-heterostructure to manufacture a H_2_ gas sensor. Graphene was deposited on Cu foil by chemical vapor deposition and transferred to Si substrate by PMMA technology [8]. A sol–gel method was used to synthesize a zinc oxide seed layer onto the surface of the graphene, and then the hydrothermal method was used to grow zinc oxide nanostructures. The structure and composition of the sensor material were analyzed with scanning electron microscope, X-ray diffractometer, and Raman spectrometer, optical microscope and photoluminescence spectroscopy.

## 2. Materials and Methods

### 2.1. Preparation and Transfer of Graphene

The graphene was prepared by chemical vapor deposition, and it was grown on a copper foil substrate at a flow ratio of 1:2 of hydrogen to methane at 950 °C. A 5 wt.% polymethyl methacrylate(PMMA, Hongyao Co., Ltd., Tainan, Taiwan) solution was coated onto the copper foil as a support layer for the graphene. The copper foil was etched and removed with ferric chloride (Hongyao Co., Ltd., Tainan, Taiwan), and then the silicon substrate was immersed in the solution so that the graphene was adsorbed onto the silicon substrate. The PMMA that supported the graphene was dissolved and removed with acetone (Hongyao Co., Ltd.,Tainan, Taiwan) [8,9].

### 2.2. Preparation of Zinc Oxide Nanostructure

The precursor solution of the zinc oxide seed layer comprised zinc acetate and sodium hydroxide as solutes and methanol as a solvent to prepare a 1 M mixed solution. After these precursors were coated by the spin-coating method, they were baked at 200 °C for 10 min, and the coating process was repeated 4 times to obtain the zinc oxide seed layer. The specimen coated with the ZnO seed layer was put into a mixed solution prepared by zinc nitrate and hexamethylenetetramine in a molar ratio of 1:1. The concentration of the solution was 0.07 M and the temperature was 95 °C. After holding the specimen at 95 °C for a period of time, the ZnO nanostructure was obtained.

### 2.3. Measurement Method of Hydrogen Sensing

The graphene/zinc oxide nano gas sensor was placed in a closed sensing chamber. The two probe heads touched the two silver electrodes on the surface of the sensor, and the other end of the probes were connected to an electrochemical workstation (Jiehan 5000, Jiehan Technology corporation, Tauchung, Taiwan) to detect the resistance variation. After the sensor was heated to the measuring temperature, different concentrations of H_2_ were introduced into the chamber and the resistance variation of the sensor was recorded. The resistance value of the sensor in the air is named R_air,_ and the resistance value of the H_2_ is named R_gas_. The sensitivity was calculated by the formula S = R_air_/R_gas_. The measurement was repeated for ten cycles under the same conditions and we checked the reproducibility is consistent.

## 3. Results and Discussion

### 3.1. Graphene

Graphene was grown on copper foil by chemical vapor deposition. By adjusting the process parameters, including the reaction temperature, reaction time, methane flow rate, and hydrogen flow rate, high-quality graphene could be obtained. The optimal reaction conditions for the deposition of graphene are a reaction temperature of 1025 °C, a reaction time of 20 min, a methane gas flow rate of 20 sccm, and a hydrogen gas flow rate of 10 sccm. The graphene grown on copper foil was transferred to a silicon substrate using PMMA, and a series of material property analyses were performed.

Figure 1 is an image of the graphene under an optical microscope; black traces were found on the surface. We speculated that these traces were a result of the copper foil surface polishing when preparing the copper substrate before the deposition. According to the nucleation theory, the dent on the surface of the substrate is the easiest nucleation site, and it is easy to start the nuclease growth from the initial graphene. Figure 2 shows the Raman spectrum of the graphene analyzed with a green laser, with a wavelength of 532 nm as the detection light source. Figure 2 shows three peaks, in which the D-band (1350 cm^−1^) represents the degree of graphene defects and the G-band (1580 cm^−1^) represents the degree of graphitization of the material (carbon materials bonded with sp^2^). The 2D-band (2680 cm^−1^) indicates the degree of graphite stacking. It is generally believed that a ratio of I_2D_/I_G_ greater than 1.0 can be regarded as a single layer of graphene. The results of this experiment show an I_2D_/I_G_ ratio of 1.42, which means that the prepared graphene had a single-layer structure.

In order to analyze the visible light transmittance of graphene, the graphene was transferred to a glass substrate for UV–visible light spectrometer transmittance analysis. According to reports in the literature, the absorbance of a single layer of graphene for visible light is only 2.3% [10]. In Figure 3, it can be seen that at a wavelength of 550 nm, the light transmittance of single-layer graphene is 95.9%, so the visible light absorbance was estimated to be 4.1%. This value is greater than the theoretical absorbance value of single-layer graphene. As mentioned earlier, the undulations on the surface of the copper foil caused graphene stacking and defects during the transfer process, which affected the absorbance to a certain extent, but the absorbance was still between that of single-layer (2.3%) and double-layer graphene (4.6%). Therefore, it was inferred that the graphene had a single-layer structure.

### 3.2. Zinc Oxide Nanostructure

The zinc oxide nanostructure was grown on the zinc oxide seed layer by a hydrothermal method. The precursor of the solution was a 50 mL homogeneous aqueous solution prepared in equal volume ratios of zinc nitrate (0.07 M) and HMTA. The hydrothermal growth temperature was fixed at 95 °C, and the reaction was carried out in a constant temperature water tank. Ashfold et al. proposed a chemical reaction formula for the hydrothermal growth of ZnO nanostructures, as shown below [11]:C_6_H_12_N_4(s)_ + 6H_2_O_(l)_ → 6HCHO_(aq)_ + 4NH_3(g)_(1)
NH_3(g)_ + H_2_O_(l)_ → NH_4_^+^_(aq)_ + OH^−^_aq)_(2)
Zn^2+^_(aq)_ + 4OH^−^_(aq)_ → Zn(OH)_4_^2−^_(aq)_(3)
Zn(OH)_4_^2−^_(aq)_ → ZnO_(s)_(4)

It can be known from the above chemical reaction formula that HMTA can dissociate NH^4+^ and OH^−^ ions when dissolved in water. The decrease in the ratio of Zn^2+^/OH^−^ makes it easy to grow a zinc oxide nanostructure with a larger length and width aspect ratio. Through the control of the hydrothermal reaction time, zinc oxide nanostructures with different lengths can be grown. For the application of gas sensors, the growth of high-density nanostructures with large aspect ratios per unit area is beneficial to improve the sensitivity of zinc oxide nanostructured gas sensors.

The surface morphology of zinc oxide nanostructures grown by the hydrothermal method was observed by SEM, and the results are shown in Figure 4. It can be seen that as the hydrothermal growth took more time, the aspect ratio of the zinc oxide nanostructure increased, and a higher-density, better uniformity nanostructure could be obtained.

Figure 5 shows the XRD patterns of zinc oxide nanostructures at different growth times (3, 6, 9, and 12 h). The XRD patterns show obvious diffraction peaks, which were 31.76°, 34.44°, 36.24°, and 47.52°, respectively. Compared to JCPDS card No. 36-1451, these diffraction peaks correspond to the (100), (002), (101), and (102) crystal planes of ZnO, respectively, indicating that the prepared zinc oxide nanostructure was a polycrystalline hexagonal wurtzite structure. The (002) characteristic diffraction peak in Figure 4 is the strongest one. This infers that the zinc oxide nanostructure growth process had a tendency to preferentially grow vertically along the C axis.

Figure 6 shows the PL spectrum of zinc oxide. In the figure, it can be observed that there are two characteristic luminous regions. The first one is the obvious luminescence peak near the ultraviolet wavelength of 378 nm, which belongs to the intrinsic luminescence of zinc oxide and is also called the near band edge emission (NBE). The second light-emitting region is approximately in the 480–600 nm wavelength range and belongs to the green light band, which is also known as the deep-level emission. According to the literature [12,13,14], it is known that the higher the density of oxygen vacancies is in a ZnO microstructure, the stronger the green light PL intensity is.

The sensitivity of metal oxide gas sensors is related to the point defects of the sensing material, especially oxygen vacancies. The oxygen vacancies in the crystal structure can be used as preferential adsorption sites for reducing gases [15,16,17]. When reducing gas molecules are adsorbed on the surface of the material, they interact with point defects. This reaction formula is shown in Equation (5).

R + O_o_^x^→ RO + V_o_^.^ + e^−^(5)

Among them, O_o_^x^ is the oxygen atom located in the zinc oxide lattice, and V_o_^.^ is the oxygen vacancy. From Equation (5), it can be observed that gas molecules bind tightly with oxygen vacancies, and oxygen vacancies act as donors and release free electrons [16]. Compared to a defect-free zinc oxide surface, a zinc oxide surface with oxygen vacancies can attract more charges, thereby reducing the energy barrier and increasing conductivity. Therefore, the existence of defects in ZnO has been proved to be beneficial for gas detection. Compared to a zinc oxide surface without oxygen vacancy defects, a surface with oxygen vacancies will generate more electrons due to the adsorption of gas molecules, thus reducing the energy barrier and increasing the concentration of electrons. From the perspective of gas sensor performance, zinc oxide as the sensor material will change the electrical resistance due to the adsorption of the target gas, which is beneficial for improving the sensitivity of gas sensors.

### 3.3. Graphene/Zinc Oxide Nano-Heterostructure

The sensitivity of the graphene/zinc oxide nano-heterostructure sensor to different H_2_ concentrations was measured at 250 °C. The measured H_2_ concentrations were 5 ppm, 500 ppm, 10,000 ppm, and 150,000 ppm, and the sensing results are shown in Figure 7. The sensitivities obtained were 1.06, 1.10, 1.17, and 1.49, respectively, with the hydrogen concentrations from 5 ppm to 150,000 ppm. Figure 8 is a graph showing the change in the sensor sensitivity versus the H_2_ concentrations. The results show that as the concentration of H_2_ increased, the sensitivity of the sensor also increased. In the case of high hydrogen concentrations, the sensing sensitivity greatly increased compared to low concentrations.

In order to investigate the reproducibility of the graphene/zinc oxide nano-heterogeneous gas sensor prepared in this study, the sensor was placed in environments of different H_2_ concentrations and a fixed sensing temperature of 250 °C. The sensors continuously performed ten cycles of hydrogen sensing tests and the results are shown in Figure 9. The results show that the graphene/zinc oxide nano-heterostructure almost had the same sensing sensitivity performance under the test conditions of multiple cycles, indicating that the sensor has good reproducibility.

According to the literature [18], it is known that graphene in the air can be doped by water vapor to exhibit p-type conductivity. Undoped zinc oxide, due to the negative charge compensation effect in the structure of oxygen vacancies, mainly conducts a current with electrons and presents an n-type semiconductor material [19]. Therefore, it is speculated that one of the reasons for the improved sensing sensitivity of this gas sensor is the P–N nano-heterostructure formed by the n-type zinc oxide and p-type graphene. The I–V curve of this nano-heterojunction was measured and the results are shown in Figure 10. It can be observed that there were obvious rectification characteristics of the P–N heterojunction. When n-type zinc oxide and a p-type graphene semiconductor are in contact, in order to balance the Fermi level, energy bending will occur. Electrons and holes will combine at the junction to produce an electron depletion layer. This electron depletion layer will change due to the sensor in different sensing atmospheres. When the sensor is in the air, oxygen molecules will combine with electrons on the surface of the zinc oxide to form an electron depletion layer on the surface. At this time, the conductivity of the zinc oxide decreases, and the Fermi level also decreases. The electron depletion layer at the junction becomes wider and the overall resistance increases. When the sensor is exposed to the reducing gas, such as H_2_, the oxygen molecules originally adsorbed on the surface will be reduced, and the electrons will be released back into the zinc oxide. The conductivity of the zinc oxide rises, and the Fermi level also rises. The electron depletion layer at the junction becomes narrower and the overall resistance decreases. The change due to the rectification of the P–N junction can increase the resistance variation between the sensor in an air atmosphere and in the target gas, thereby improving the overall sensing performance.

## 4. Conclusions

In this study, single-layer graphene with I_2D_/I_G_ = 1.43 was prepared by chemical vapor deposition and transferred to a silicon substrate by PMMA transfer technology. One-dimensional zinc oxide nanostructures were prepared by spin-coating and hydrothermal methods and grown on the graphene top surface. The obtained graphene/zinc oxide nano-heterostructure gas sensors had good sensitivity in the hydrogen concentration range of 5 ppm to 150,000 ppm, with good sensitivity and repeatability. This can mainly be attributed to the high specific surface area of nanostructured zinc oxide and the P–N heterojunction formed between zinc oxide and graphene.

## Figures and Tables

**Figure 1 materials-14-06943-f001:**
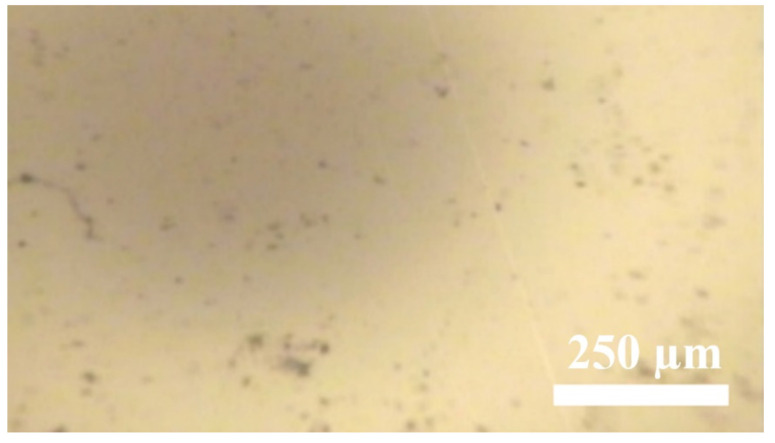
Optical microscope image of graphene on silicon substrate.

**Figure 2 materials-14-06943-f002:**
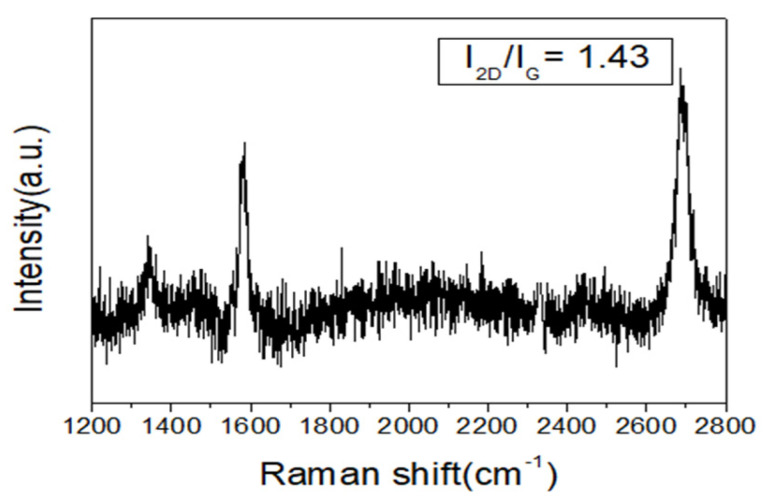
Raman spectrum of graphene on silicon substrate.

**Figure 3 materials-14-06943-f003:**
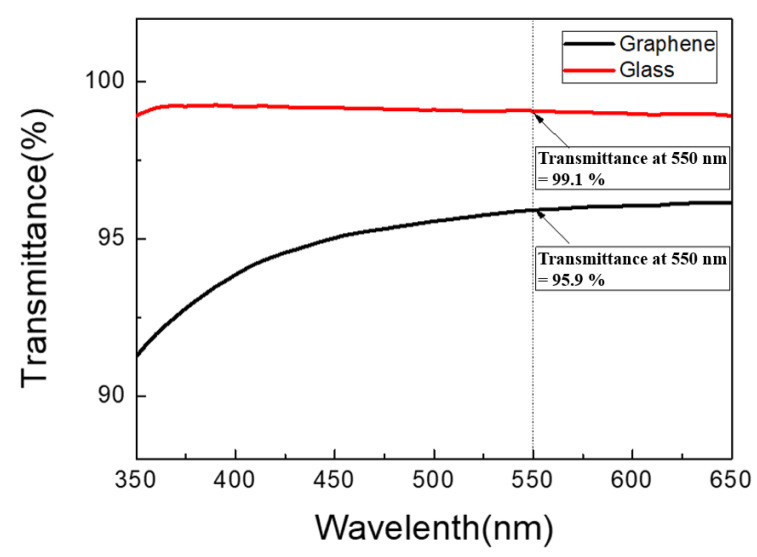
UV–visible transmittance spectrum of graphene.

**Figure 4 materials-14-06943-f004:**
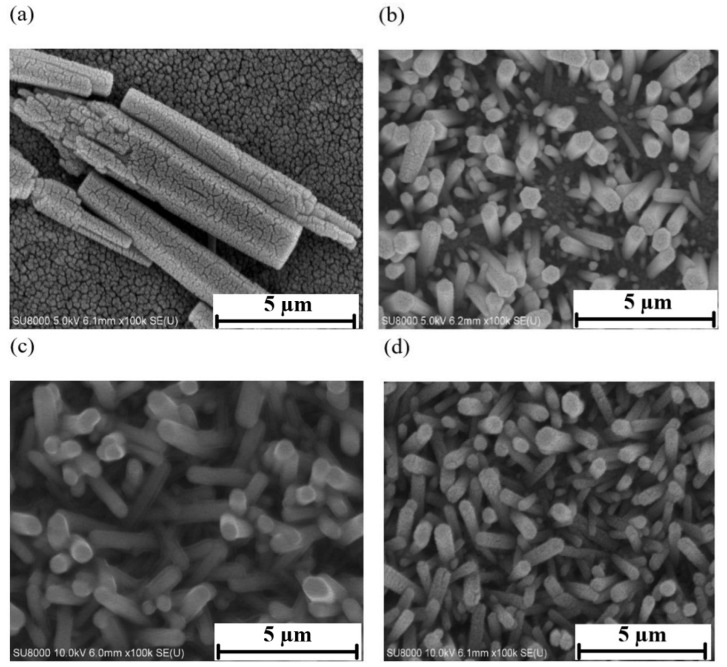
Hydrothermal growth of zinc oxide nanostructures. (**a**) 3 h (**b**) 6 h (**c**) 9 h (**d**) 12 h.

**Figure 5 materials-14-06943-f005:**
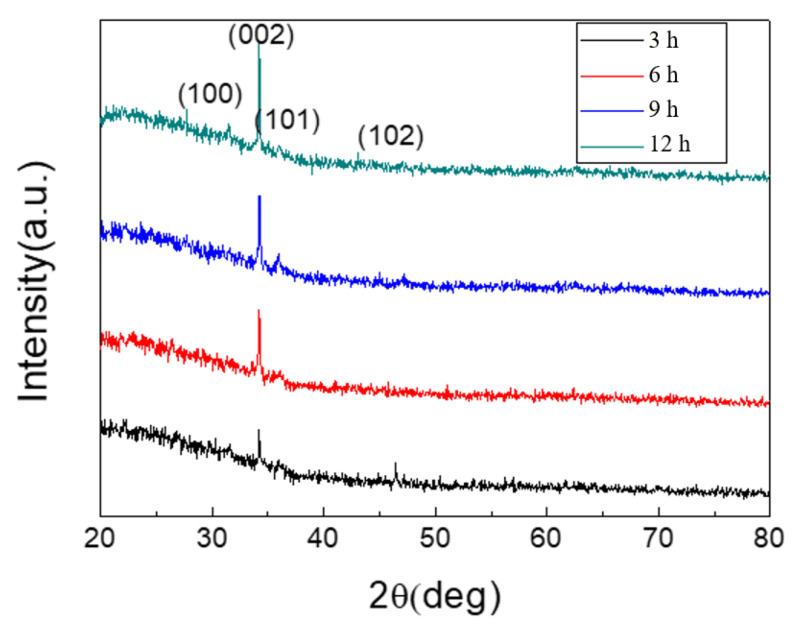
XRD patterns of zinc oxide nanostructures at different growth times.

**Figure 6 materials-14-06943-f006:**
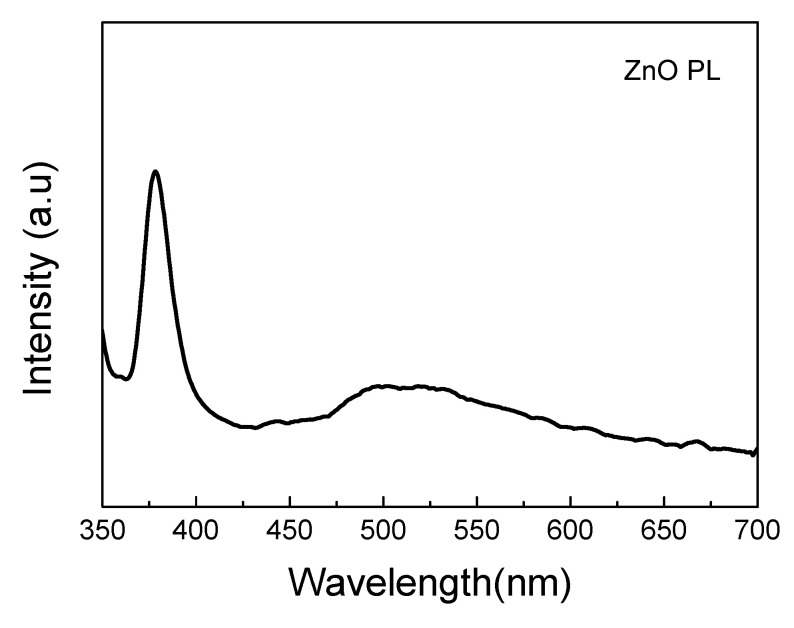
The PL spectrum of zinc oxide nanostructure.

**Figure 7 materials-14-06943-f007:**
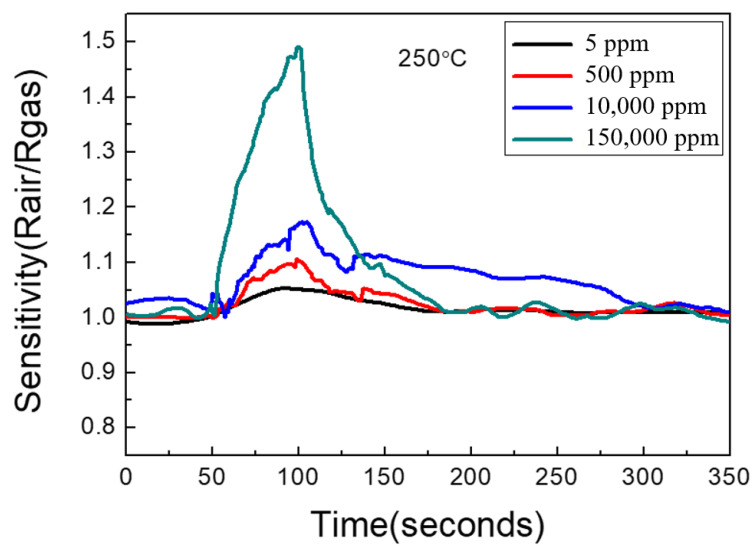
Sensitivity measurements of graphene/zinc oxide nano-heterostructures to different H_2_ concentrations at 250 °C.

**Figure 8 materials-14-06943-f008:**
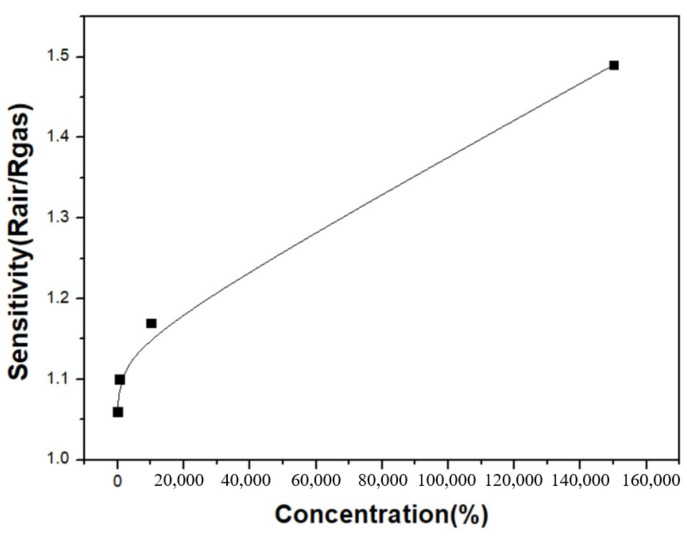
The sensitivity changes in the ZnO/graphene heterojunction sensor versus the variation in the H_2_ concentrations at 250 °C.

**Figure 9 materials-14-06943-f009:**
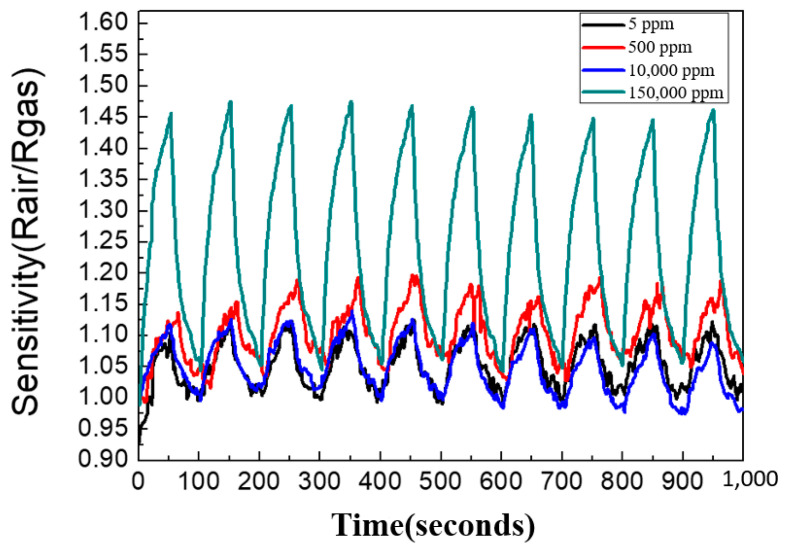
The reproducibility test of the graphene/zinc oxide nano-heterostructure at 250 °C with different H_2_ concentrations.

**Figure 10 materials-14-06943-f010:**
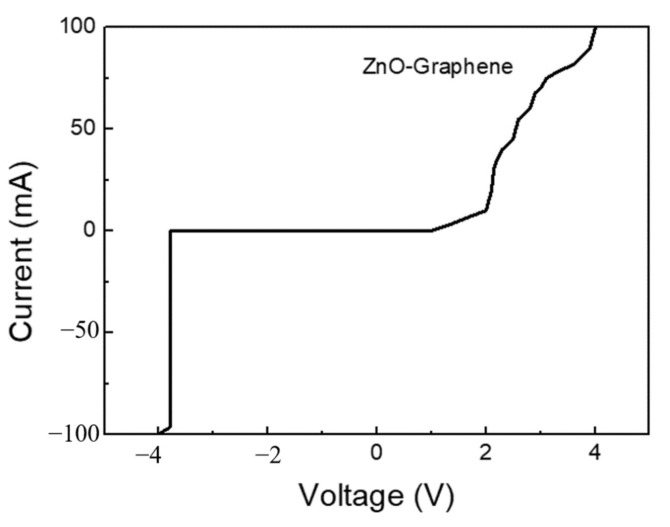
The I–V electrical character of graphene/zinc oxide nano-heterostructure.

## Data Availability

The data presented in this study are available on request from the corresponding author.

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
