# Peer review of "Fabrication of Graphene/Zinc Oxide Nano-Heterostructure for Hydrogen Sensing"

_materials, 2021, doi:10.3390/ma14226943_

Round 1
Reviewer 1 Report
The manuscript falls on the scope and goal of the journal. There is novelty of the work, however require some addition and modification for better understanding.
- Add a reference of PMMA transfer technology.
- Details and more references need to add in introduction section.
- To confirm the thin layer of graphene growth, X-RD and d-spacing are necessary to add
- Fig-1. Please add a clear TEM image
- Fig-5 Image scale is not clear- please add the image scale clearly
- Line 94-105, please add references to justify your findings.
- conclusion must be more accurate and justified.
Author Response
Response to Reviewer 1 Comments
Point 1:Add a reference of PMMA transfer technology.
Response 1:
We had added reference 8 and reference 9 for graphene transfer as shown below:
[8]Sami Ullah, et al. Graphene transfer methods: A review, Nano Research 2021,14,3576-3772
[9]Jen, Shuo-Fang.; Chiu Po-Wen. Cu-catalyst graphene synthesis by chemical vapor deposition and its electrical properties of ammonia-doped graphene nanoribbons. 2011, DOI: 10.6843/NTHU.2011.00559
Point 2: Details and more references need to add in introduction section.
Response 2: We had added our previous work about the nano-ZnO hydrogen sensing as reference[7]. Details and more references could be found in the references in that paper.
Point 3: To confirm the thin layer of graphene growth, X-RD and d-spacing are necessary to add
Response 3: We showed the Raman spectrum of graphene in Fig. 2, which can more clearly determine that the growing film is graphene instead of graphite or carbon. Raman spectrometer is more effective than XRD in analyzing graphene. XRD cannot distinguish these three different types of carbon materials. So there is no need to add XRD data for graphene.
Point 4: Fig-1. Please add a clear TEM image
Response 4: Fig.1 is just to show the distribution of graphene deposited on the copper substrate, Therefore, observation with an optical microscope is sufficient, and it is not necessary to observe with TEM.
Point 5: Fig-5 Image scale is not clear- please add the image scale clearly
Response 5 : We’ve re-labeled the scale bar and it’s clear now.
Point 6: Line 94-105, please add references to justify your findings.
Response 6 :We had added more references in the content.
Point 7: conclusion must be more accurate and justified.
Response 7 :We added the following sentences in the conclusion:“This nano heterostructure can detect hydrogen concentration as low as 5ppm, which is a great improvement.” “This results demonstrate the technical feasibility of the comprehensive application of nanomaterials and p-n heterojunction in hydrogen sensing.”

Reviewer 2 Report
In this work, the authors prepared graphene/nano-zinc oxide heterostructure by growing ZnO nanorods (using hydrothermal) on a single layer graphene and used this structure for hydrogen sensing. The obtained heterostructure showed a sensitivity of 1.06 at a sensing temperature of 205 °C and a concentration of hydrogen as low as 5 ppm. The idea of preparing graphene/nano-zinc oxide heterostructure and using them as sensors has already been reported in several works. So I find the importance of this paper mainly in the possible sensing of H2 gas by this structure. The advancement that has been made is fair to warrant publication in Materials after major revisions.
-In the introduction, the authors should also cite the pioneering works that started this area of research.
-The authors need to determine the dimensions of ZnO by determining their size distributions and check also the formation of ZnO nanoparticles. It is not clear how this will affect the reproducibility of the measured sensitivity and other properties?
-The reported SEM images are unclear and not enough to claim the formation of only ZnO nanorods. High resolution SEM images should also be included.
- The authors should also report some of the sensing data between 10000 and 150000 pm to check how it varies.
- How the nanorod structure and spherical ZnO nanoparticles grown on a single graphene layer are compared when applied in the H2 sensing?
-I have also reservations about some of the authors’ conclusions and assumptions about the factors that contributes to hydrogen sensing because there are not enough evidences.
Author Response
Response to Reviewer 2 Comments
Point 1:In the introduction, the authors should also cite the pioneering works that started this area of research.
Response 1: We had cited our previous research results as reference[7].The sentences we added in the introduction are as follows: “In our previous research results, we have reported the influence of ZnO seed layer on hydrogen sensing in ZnO nanostructures[7].”
Point 2:The authors need to determine the dimensions of ZnO by determining their size distributions and check also the formation of ZnO nanoparticles. It is not clear how this will affect the reproducibility of the measured sensitivity and other properties?
Response 2: Growing ZnO nanorods by hydrothermal method is a very mature technology in our laboratory. We use the best conditions to grow ZnO nanorods. Basically, the length of the zinc oxide nanorods increases with the growth time. If the growth time was fixed, the length difference will not be too large. When the growth time is short, short columns are formed instead of granular.
Point 3:The reported SEM images are unclear and not enough to claim the formation of only ZnO nanorods. High resolution SEM images should also be included.
Response 3: Growing ZnO nanorods by hydrothermal method is a very mature technology in our laboratory. We use the best conditions to grow ZnO nanorods. Because ZnO is semiconductive, it cannot be as clear as conductive materials when observed by SEM. The provided SEM photos in this study should be clear enough to see the nanorod structures.
Point 4: The authors should also report some of the sensing data between 10000 and 150000 pm to check how it varies.
Response 4:The important thing for hydrogen sensing is to be able to detect low concentrations of hydrogen. The sensing of hydrogen greater than 10000ppm does not have much practical significance.
Point 5: How the nanorod structure and spherical ZnO nanoparticles grown on a single graphene layer are compared when applied in the H2 sensing?
Response 5:In this study, we use the hydrothermal method to grow zinc oxide nanorods. There is no spherical zinc oxide formation. If both of the two are used for hydrogen sensing, it can be expected that the zinc oxide nanorod has a larger surface area and more conducive to hydrogen sensing.
Point 6:I have also reservations about some of the authors’ conclusions and assumptions about the factors that contributes to hydrogen sensing because there are not enough evidences.
Response 6: Since this paper focuses on showing the combination of ZnO nanostructure and Graphene/ZnO nanoheterostructure, it can be applied to low-concentration hydrogen sensing. More detailed researches are underway. Actually we are conducting research on the correlation between heterogeneous nanostructures and ZnO point defects for hydrogen sensing. After these results are completed, they will be compiled into another paper for publication.
